# How does social desirability bias influence survey-based estimates of the use of antenatal care in rural Nepal? A validation study

Andrew L Thorne-Lyman [ID],[1] Tsering P Lama,[1,2] Rebecca A Heidkamp,[1]
Melinda K Munos [ID],[1] Porcia Manandhar,[1] Subarna K Khatry,[1,2] Emily Bryce,[1,3]
Steven C LeClerq,[1,2] Joanne Katz[1]

[1]Department of International Health, Johns Hopkins University Bloomberg School of Public Health, Baltimore, Maryland, USA
[2]Nepal Nutrition Intervention Project, Sarlahi, Kathmandu, Nepal
[3]Jhpiego, Baltimore, MD, USA

**Correspondence to**
Dr Andrew L Thorne-Lyman; athorne1@jhu.edu

## ABSTRACT

**Objectives** Social desirability bias is often speculated to influence survey responses but seldom studied in healthcare. The objective was to explore whether social desirability scores (SDS) or the presence of interview observers is associated with inaccurate recall and overestimation of antenatal care (ANC) services.

**Design** Longitudinal validation study comparing recalled receipt of ANC services and nutrition components of ANC against direct observations of care. An adapted short form Marlowe-Crowne questionnaire was used to generate an SDS, and the presence of interview observers was treated as a separate exposure. We assessed accuracy and overestimation of recalled receipt of ANC services against observed receipt using log-binomial regression, adjusting for age, education, first-pregnancy and socioeconomic status.

**Setting** Rural Southern Nepal with recruitment from five government health posts.

**Participants** 401 pregnant women.

**Results** Social desirability scores did not significantly predict accuracy or overestimation of most types of ANC care except counselling on nausea. Higher SDS was associated with more accurate recall (adjusted RR, aRR 1.08 (95% CI 1.03, 1.12)) and less overestimation (aRR 0.85 (0.80, 0.91)). The presence of mothers-in-law or husbands during interviews was associated with greater overestimation of the number of ANC visits received by more than three visits (aRR 2.07 (1.11, 3.84)) and (aRR 4.19 (2.17, 8.10)), respectively. Those interviewed with friends present tended to overestimate the receipt of counselling on nausea, avoiding alcohol and not smoking.

**Conclusion** The presence of observers can lead to overestimation of the receipt of ANC care and support the conduct of interviews in private settings despite challenges of doing so in village contexts. Findings that the SDS did not predict the accuracy of most types of ANC care might reflect a reality that such questions may not be sensitive from a social-norms perspective. Additional local adaptation of SDS is recommended.

## INTRODUCTION

The WHO recommends a number of nutrition interventions in pregnancy, including

## STRENGTHS AND LIMITATIONS OF THE STUDY

⇒ The effect of social desirability bias and the presence of interview observers on response accuracy was evaluated against the gold standard of observed receipt of antenatal care services.
⇒ The tool used to assess social desirability was adapted for comprehension for the local population in rural Nepal but has not been validated for that context.
⇒ Interviewers noted the presence of others at any point during the interview, and not specifically during specific portions of the interview, likely attenuating estimates of the relationship between the presence of others and validity of reported services receipt.

iron-folic acid (IFA) supplements, counselling on healthy eating and physical activity, and in certain contexts calcium, micronutrient supplements and balanced energy and protein supplements.[1] Large-scale household surveys, including those implemented by the Demographic Health Survey Programme, are used across low-income and middle-income countries to measure coverage of antenatal interventions in order to monitor progress towards targets and evaluate policies and programmes.[2]

Many survey-based coverage indicators come from self-reported data, and a growing number of studies have explored the validity of responses to questions about the receipt of nutrition interventions delivered through antenatal care (ANC).[2 3] A study carried out by our research team in Nepal found that respondents tend to overestimate the receipt of IFA in pregnancy and that overestimation was not explained by either maternal or household characteristics.[4] Women's reports of the receipt of various counselling interventions, weight measurement and deworming

were generally recalled more accurately than questions about other services such as counselling to not drink alcohol or tobacco.[5]

Social desirability bias (SDB) is a common type of response bias that reflects the tendency of respondents to reply in a manner that may be viewed positively by social peers or that is consistent with social norms and expectations. SDB is generally thought to be greater for questions that are perceived as sensitive or subject to judgement such as those related to sexual practices or family planning.[6 7] The perceived relationships between respondents and interviewers can also influence socially desirable responses. For example, a study administered in 14 African countries found that respondents give systematically different answers to interviewers depending on whether or not they were from their same ethnic group.[8]

Survey-based scales to measure social desirability were developed in the field of social psychology in the 1960s beginning with the 33-question Marlowe-Crowne scale.[9] Shorter versions of these scales have been shown to be reliable across different cultural contexts, though more work is needed to refine and test them.[7 10 11] These scales include a variety of positively keyed questions that are 'culturally acceptable but improbable' or 'desirable but rare' and negatively keyed items that are 'culturally unacceptable but probable' or 'undesirable but common'.[10] The scales are used to detect and quantify the extent of socially desirable responses in surveys. Where it has been detected, a number of approaches exist correct for it or mitigate its effects including a posteriori statistical adjustment for social desirability scale (SDS),[12 13] excluding high SDS scorers[13] or rephrasing questions using forgiving wording.[14]

SDB has been frequently described as an area of potential concern in nutrition-related studies collecting data about diet, physical activity and self-reported weight, but few studies have directly quantified the strength of association between social desirability scores and these domains.[15] Evaluations of the Alive and Thrive infant and young child-feeding programme in Vietnam and Bangladesh explored whether adjusting for short-form SDSs affected estimates of programme impact on diet but found that results were unchanged.[16 17] To our knowledge, no previous studies have directly explored whether SDB or the presence of others during survey interviews is a potential explanation for inaccurate survey responses about the receipt of ANC and related nutrition interventions. The aims of this study were therefore to:

1. Adapt and test a modified short-form version of the Marlow-Crowne SDS in rural Nepal.
2. Explore whether social desirability or the presence of other adults during interviews helps to explain previous findings related to overestimation of the receipt of services.
3. Explore whether social desirability or the presence of other adults during interviews is associated with inaccurate recall of ANC and associated services.

## METHODS

### Study area and population

This study took place in Sarlahi District, an area in the southern plains (*Tarai*) of Nepal located in Province 2, the province with the lowest rates of ANC coverage in Nepal. According to the 2016 Demographic and Health Survey, only 36% of the women in Province 2 received all four recommended ANC visits, far below the national average of 59%.[18] These low rates appear paradoxical given that 70% of the population in Sarlahi lived within 30 min of a government health facility. This paradox may be explained in part by low rates of maternal education and gender norms that limit women's ability to travel outside of the household.[19] The population of Sarlahi is linguistically and culturally heterogeneous, including both Nepali-speaking and Maithili-speaking groups.[20]

### ANC and nutrition services delivered through ANC

At the time of our study, Nepal was implementing the WHO's focused ANC model, with recommended visits occurring in the 4, 6, 8 and 9 months of pregnancy. The government incentivises ANC participation by providing 800 rupees (~7 USD) to women who complete all four visits.[21] Nutrition interventions delivered through ANC in Nepal include the provision of IFA supplements, checking of weight and counselling on various topics related to nutrition and breast feeding.

### Study design

This study was a part of a larger effort to validate coverage indicators for various nutrition services through ANC, described in detail elsewhere.[4 5] In brief, study staff were based at five government health posts and trained to directly observe and record the receipt of ANC services by enrolled women. The observers' 28-item checklist included details about the provision of IFA pills (eg, how many pills women received, whether they were instructed to purchase IFA tablets or syrup), measurement of weight, and types of counselling on nutrition and breast feeding. These direct observations served as a 'gold standard' to validate maternal recall of ANC services received at study clinics. Maternal recall was assessed by interviewing enrolled women using a structured questionnaire around 6 months after delivery.[2 4 5] The interview questions were drawn from the official translated Nepali DHS 2016 questionnaire (DHS Round 7), a draft of the DHS Round 8 questionnaire and novel questions to assess different components of counselling and weight gain assessment (online supplemental table 1). The 6-month visits were conducted at the women's homes or their parental homes. Women were asked about socioeconomic status at enrolment to the validation study and during the visit at 6 months after delivery. As described in detail elsewhere,[4] a target sample size of 300 women was established for the study based on an assumption of 50% coverage for receipt of IFA and lower coverage for counselling. After accounting for loss-to-follow-up, the possibility of women

going elsewhere for ANC, and of non-live birth outcomes, a target study enrolment of 450 women was established.

## Measurement of social desirability

Questions from a short-form Marlowe-Crowne social desirability index (online supplemental table 2) were translated and back-translated by Nepali/English speakers familiar with the study context, pretested and slightly adapted to the local context. During pretesting, one of the questions was not well understood and dropped; 12 questions were included in the 6-month visit questionnaire.[11] Study staff also recorded whether any other adult had been present at any point during the interview (with specific codes for husbands, mothers-in-law (MILs), friends of the respondent or other adults), as well as whether those present had helped the respondent to answer any questions.

Correlations between different items of the social desirability index were explored after reverse coding negatively keyed questions, and most were found to be positively associated with one another (online supplemental table 3). Tests of internal consistency revealed the full 12-item scale to have a Cronbach's alpha of 0.58, and only minor improvements (to an alpha of 0.65) were possible by removing several items, so the full 12-item scale was used in the analyses. On visual inspection, scores on the SDS were only slightly skewed, with a mean of 5.5 and a SD of 2.1.

## Data collection

Enrolment criteria for the parent study included currently pregnant, married women aged 15 years or older, who (a) resided in the study area, (b) visited one of the five health posts for their first ANC visit and (c) had the intention to return to these study sites for subsequent ANC visits. All interview questions were translated and back-translated by Nepali/English speakers familiar with the study context. These questions were pretested, and minor changes were made to help facilitate comprehension in the local context. Both Maithili-speaking and Nepali-speaking interviewers conducted the interviews following training exercises designed to standardise the approach to translation. Interviewers were trained to replicate usual survey conditions. This included asking respondents to answer questions themselves even if a family member had tried to answer it for her and repeating or rephrasing questions in a simpler way if respondents were confused or responded with 'I don't know'.

The study started in December 2018 and reached full enrolment by November 2019. Direct observations of ANC visits were nearly completed by March 2020 when a shutdown for the COVID-19 pandemic interrupted all non-emergency health services in Nepal. Data collection for the postpartum 6-month visit was started in September 2019, but COVID-19-related shutdowns caused delays. The mean number of days that passed from delivery to the postpartum data collection visit was 252 with an SD of 75 days. Precautions taken to prevent the spread of COVID-19 included masking by interviewers and respondents, handwashing with soap provided to interviewees, holding interviews outdoors when weather permitted and maintaining a distance of more than 6 feet during data collection.

## Data analysis

We examined whether SDS or the presence of other adults during interviews was associated with the accuracy or overestimation of recalled receipt of specific ANC services using log-binomial bivariate and multivariable regressions to directly estimate risk ratios or log-Poisson models in the case of model non-convergence.[22] Multivariable analyses adjusted for age (continuous), education (any vs none), first pregnancy (yes/no) and quartile of socioeconomic status as possible confounders. We used a theory-based approach to selection of covariates for these models. For interpretation purposes, high SDS corresponds to women who would be more likely to exhibit socially desirable responses. SDS was treated as both a continuous score and as a dichotomous variable in analyses, with the upper quartile defined as those with a score of 7 or greater compared against the lower three quartiles, recognising that socially desirable response bias may not operate in a linear manner. For receipt of counselling interventions, calcium and deworming, accurate recall was defined as alignment (yes/no) of the follow-up interview responses with observed receipt or non-receipt of those interventions.[4 5] Overestimation for these interventions was defined as the percent of respondents who reported receipt but were not observed receiving it, and for each intervention, women were categorised as over estimators or non-over estimators. For ANC visits and IFA, several different thresholds were used to define accuracy and the degree of overestimation. For reasons of statistical power, we restricted our analysis to questions about interventions with directly observed coverage below 90%. A complete case analysis was used for the treatment of missing data, and loss to follow-up was minimal (online supplemental figure 1).

## Institutional Review Board approval

This research conformed with the principles embodied in the Declaration of Helsinki. The institutional review boards at Johns Hopkins Bloomberg School of Public Health and the Nepal Health Research Council both approved this study including permission to collect and resume data collection during the COVID-19 epidemic. Signed consent was obtained from all participants. Patients and the public were not involved in the design, conduct, reporting or dissemination plans of this research. Dissemination and discussion of study design and results are made to a national committee and a local district committee.

## Patient and public involvement

The research questions were not informed through direct consultation of patients' priorities, experience or preferences. However, many members of our local research

| Table 1 | Characteristics of the study population | | |
|---|---|---|---|
| | Bottom three quartiles SDS | Upper quartile SDS | Overall |
| | (n=284) | (n=112) | (n=396) |
| Age, years and mean (SD) | 22.7 (4.3) | 21.9 (4.0) | 22.5 (4.2) |
| # Observed antenatal care visits, mean (SD) | 4.6 (2.4) | 4.8 (2.6) | 4.7 (2.5) |
| Socioeconomic status quartile (%) | | | |
| 1 (Bottom) | 36.6 | 39.3 | 37.4 |
| 2 | 17.6 | 17.0 | 17.4 |
| 3 | 31.7 | 29.5 | 31.1 |
| 4 (Top) | 14.1 | 14.3 | 14.1 |
| Any prior live births (%) | 71.7 | 71.8 | 71.7 |
| Any formal education (%) | 32.2 | 22.3 | 28.3 |
| Trimester at enrolment (%) | | | |
| 1–3 months | 39.4 | 50.0 | 42.4 |
| 4–6 months | 59.2 | 47.3 | 55.8 |
| 7–9 months | 1.4 | 2.7 | 1.8 |
| Interview observed by another adult (%) | 60.9 | 31.1 | 52.1 |
| Husband | 7.4 | 5.9 | 7.0 |
| Mother-in-law | 28.3 | 14.3 | 24.4 |
| Friend | 15.9 | 5.0 | 12.9 |
| Other adult | 39.5 | 16.0 | 32.9 |
| Observer helped respondent answer questions (%) | 29.0 | 23.5 | 27.9 |

SDS, social desirability scale.

team are from the local population where the research is conducted and were involved in pretesting, questionnaire development and recruitment. Findings will be disseminated to national and local stakeholders through meetings and local presentations.

## RESULTS

Of the 441 women enrolled in the study, 396 were included in our analysis after accounting for exclusion criteria described in online supplemental figure 1. The mean age, number of observed ANC visits and birth history were similar among women in the upper quartile of SDS compared with those in the lower three quartiles (table 1). Women with lower SDS tended to have lower education and enrolled in the study later in pregnancy than those with higher scores. Twice as many women with lower SDS had another adult present during the interview than those with higher scores; MILs and other adults were the most common observers. Those with lower SDS were only slightly more likely to have

an observer help them respond to interview questions. Characteristics of women disaggregated by the presence of observers are presented in online supplemental table 4; those with a friend present were less likely to be in the top quartile of socioeconomic status or to have education.

About half of women overestimated the number of IFA tablets they received by at least 30 tablets and over a third overestimated the number of ANC visits by at least one visit (table 2). About a third of women reported receiving counselling for nausea, alcohol or smoking that had not been observed at clinics. Null relations were apparent between SDS and overestimation of receipt of most services, although some associations neared statistical significance when SDS was treated as a dichotomous predictor. Additionally, the dichotomous SDS had a stronger relation with extreme overestimation of reported receipt of both ANC and IFA compared with more modest overestimation. SDS were generally not associated with accuracy of recalled services (table 3). The only exception was for counselling on the management of nausea: women with higher SDS tended to have more accurate recall.

The presence of any other adult was significantly associated with overestimation of receipt of counselling on managing nausea as well as overestimation of the number of ANC visits. MILs were present during about a quarter of all interviews. Their presence was associated with significant overestimation of the receipt of deworming tablets (ARR 1.90 (95% C.I. 1.18, 3.05)) and more than doubled the risk of overestimating the number of ANC visits by three or more visits (ARR 2.07 (95% 1.11, 3.84)) (table 4). Interestingly, the presence of an MIL during the interview appeared to attenuate overestimation of IFA receipt by >90 tablets, though this estimate included the null (ARR: 0.56 (95% CI 0.31, 1.01)). The presence of a husband was less common (7%) and was significantly associated with the overestimation of ANC visits by one, two and three visits. The presence of a friend was significantly associated with an overestimation of the receipt of multiple types of ANC services including information about calcium, managing nausea, not drinking alcohol, and not smoking or using tobacco.

Women interviewed in the presence of MILs were less accurate in estimating the number of ANC visits but more accurate in recalling the receipt of counselling related to avoiding alcohol consumption in pregnancy. Women tended to be less accurate when estimating the number of ANC visits or receipt of IFA in front of their husbands (table 5).

We also examined how the receipt of help from observers to answer questions during the interview may have affected the accuracy of recall of services (online supplemental table 5). Assistance from MIL was associated with more accurate recall of counselling to not drink alcohol (ARR 1.24 (95% CI 1.00, 1.55)) but no other services. In contrast, assistance from husbands did not significantly affect the accuracy of recall of any services.

**Table 2** High SDS as a predictor of overestimation of nutrition/ANC services receipt in pregnancy

| | | | SDS* as a predictor of overestimation | |
| | N | Proportion overestimating % | Dichotomous SDS adjusted†‡ RR (95% CI) | Continuous SDS adjusted RR†‡ (95% CI) |
|---|---|---|---|---|
| Deworming | 395 | 15.2 | 1.12 (0.67, 1.87) | 1.01 (0.90, 1.13) |
| Receipt or prescribe Calcium | 390 | 7.6 | 0.76 (0.33, 1.75) | 0.90 (0.75, 1.07) |
| Counselling on | | | | |
| Managing nausea | 391 | 38.3 | 0.49 (0.34, 0.72) | 0.85 (0.80, 0.91) |
| Not drinking alcohol | 395 | 37.7 | 1.26 (0.97, 1.63) | 1.04 (0.98, 1.10) |
| Not smoking/using paan | 395 | 31.2 | 1.23 (0.91, 1.66) | 1.02 (0.95, 1.09) |
| Overestimate iron-folic acid receipt | | | | |
| By >30 tablets | 404 | 51.7 | 1.07 (0.87, 1.30) | 1.02 (0.97, 1.06) |
| By >60 tablets | 404 | 34.0 | 1.16 (0.87, 1.54) | 1.03 (0.97, 1.10) |
| By >90 tablets | 404 | 17.5 | 1.28 (0.83, 1.98) | 1.02 (0.93, 1.13) |
| Overestimate # of ANC visits | | | | |
| By >1 visit | 402 | 34.8 | 0.99 (0.74, 1.34) | 1.00 (0.94, 1.07) |
| By >2 visits | 402 | 18.7 | 1.17 (0.75, 1.82) | 1.02 (0.92, 1.13) |
| By >3 visits | 402 | 9.2 | 1.37 (0.71, 2.62) | 1.02 (0.87, 1.18) |

*SDS=social desirability score, treated as upper quartile versus lower three quartiles.
†Adjusted risk ratio >1 indicates greater risk of overestimating receipt of services.
‡Adjusted for woman's age (continuous), any education, first pregnancy and socioeconomic status quartile.
ANC, antenatal care.

## DISCUSSION

Previous studies exploring the validity of questions asking respondents to self-reported receipt of ANC services have raised SDB as a possible type of error that might affect coverage estimates.[23 24] To our knowledge, however, this is the first study in the peer-reviewed literature to directly explore whether SDB or the presence of others influences the validity of women's recalled receipt of ANC services. We found that SDS did not significantly predict the accuracy of women's recall of most types of ANC services, except for counselling on the management of nausea. However, the presence of another adult observer

**Table 3** High SDS as a predictor of accurate recall of nutrition/ANC services receipt in pregnancy

| | | | SDS* as a predictor of accurate recall | |
| | N | Proportion accurate % | Dichotomous SDS adjusted†‡ RR (95% CI) | Continuous SDS adjusted RR†‡ (95% CI) |
|---|---|---|---|---|
| Deworming | 395 | 77.5 | 0.97 (0.86, 1.09) | 0.99 (0.96, 1.01) |
| Receipt or prescribe Calcium | 390 | 85.6 | 0.98 (0.89, 1.09) | 0.99 (0.97, 1.01) |
| Counselling on | | | | |
| Managing nausea | 391 | 50.9 | 1.30 (1.07, 1.60) | 1.08 (1.03, 1.12) |
| Not drinking alcohol | 395 | 55.7 | 0.89 (0.72, 1.10) | 0.97 (0.93, 1.02) |
| Not smoking/using paan | 395 | 59.5 | 0.91 (0.74, 1.11) | 0.98 (0.94, 1.02) |
| Overestimate iron-folic acid receipt | | | | |
| By >30 tablets | 396 | 39.9 | 0.87 (0.66, 1.16) | 0.98 (0.93, 1.04) |
| By >60 tablets | 396 | 61.6 | 1.01 (0.85, 1.20) | 1.02 (0.98, 1.05) |
| By >90 tablets | 396 | 79.3 | 0.91 (0.80, 1.03) | 1.00 (0.97, 1.02) |
| Overestimate # of ANC visits | | | | |
| By >1 visit | 396 | 21.0 | 0.94 (0.60, 1.45) | 1.01 (0.92, 1.11) |
| By >2 visits | 396 | 55.1 | 0.91 (0.74, 1.12) | 1.02 (0.97, 1.06) |
| By >3 visits | 396 | 75.5 | 0.96 (0.84, 1.10) | 1.01 (0.98, 1.04) |

*SDS=social desirability score, treated as upper quartile versus lower three quartiles.
†Adjusted risk ratio >1 indicates more accurate recall.
‡Adjusted for woman's age (continuous), any education, first pregnancy and socioeconomic status quartile.
ANC, antenatal care.

**Table 4** Presence of others during the interview as a predictor of overestimation of recalled services receipt

| | N | Mother-in-law present ARR† (95% CI) | Husband present ARR† (95% CI) | Friend present ARR† (95% CI) | Any adult present ARR† (95% CI) |
|---|---|---|---|---|---|
| Deworming | 401 | 1.90 (1.18, 3.05) | 0.50 (0.13, 1.94) | 0.85 (0.41, 1.76) | 1.05 (0.66, 1.67) |
| Receipt or prescribe Calcium | 396 | 1.02 (0.45, 2.32) | 1.63 (0.52, 5.06) | 2.69 (1.30, 5.55) | 1.91 (0.92, 3.97) |
| Counselling on | | | | | |
| Managing nausea | 397 | 1.17 (0.89, 1.54) | 1.09 (0.68, 1.76) | 1.53 (1.15, 2.04) | 1.70 (1.30, 2.22) |
| Not drinking alcohol | 401 | 0.80 (0.58, 1.11) | 1.07 (0.66, 1.74) | 1.41 (1.05, 1.90) | 1.09 (0.85, 1.39) |
| Not smoking/using paan | 401 | 0.71 (0.48, 1.06) | 1.22 (0.73, 2.06) | 1.55 (1.12, 2.15) | 0.98 (0.74, 1.31) |
| Overestimate iron-folic acid receipt | | | | | |
| By >30 tablets | 402 | 0.94 (0.76, 1.17) | 0.88 (0.60, 1.29) | 0.93 (0.69, 1.25) | 0.96 (0.80, 1.15) |
| By >60 tablets | 402 | 0.86 (0.62, 1.19) | 1.13 (0.72, 1.77) | 0.89 (0.58, 1.37) | 0.95 (0.73, 1.24) |
| By >90 tablets | 402 | 0.56 (0.31, 1.01) | 0.90 (0.40, 2.02) | 0.96 (0.51, 1.81) | 0.73 (0.49, 1.11) |
| Overestimate # of antenatal care visits | | | | | |
| By >1 visit | 400 | 1.28 (0.97, 1.70) | 1.48 (1.03, 2.13) | 1.18 (0.82, 1.70) | 1.33 (1.01, 1.73) |
| By >2 visits | 400 | 1.32 (0.85, 2.05) | 1.97 (1.13, 3.44) | 1.24 (0.70, 2.20) | 1.57 (1.03, 2.40) |
| By >3 visits | 400 | 2.07 (1.11, 3.84) | 4.19 (2.17, 8.10) | 0.43 (0.10, 1.72) | 2.26 (1.15, 4.46) |

*Adjusted risk ratio >1 indicates greater risk of overestimation of reported services receipt.
†Adjusted for woman's age (continuous), any education, first pregnancy and socioeconomic status quartile.

during the interview did influence the accuracy of recall. Women who were interviewed with their MIL or husbands present tended to overestimate the number of ANC visits they had received, and those interviewed with friends present tended to overestimate the receipt of different types of counselling.

The effects of SDB are complex,[25] and socially desirable responses may be driven by either unconscious distortions related to self-deception or impression management related to social norms.[13 26] The potential for SDB to affect questions about nutrition and receipt of ANC services in low-income and middle-income countries has only been indirectly explored in the literature. For example, a study conducted in China found that recalled receipt of hepatitis B blood tests was much higher than HIV tests despite similar observed coverage and suggested that the difference might be explained by socially desirable responses.[23] Many of the questions that we tested in our study related

**Table 5** Presence of others during the interview as a predictor of accuracy of recalled services receipt

| | N | Mother-in-law present RR (95% CI) | Husband present RR (95% CI) | Friend present RR (95% CI) | Any adult present RR (95% CI) |
|---|---|---|---|---|---|
| Deworming | 401 | 0.91 (0.79, 1.04) | 1.13 (0.99, 1.29) | 0.84 (0.57, 1.25) | 1.03 (0.93, 1.14) |
| Receipt or prescribe Calcium | 396 | 1.03 (0.94, 1.12) | 0.97 (0.81, 1.16) | 0.87 (0.74, 1.04) | 0.99 (0.92, 1.08) |
| Counselling on | | | | | |
| Managing nausea | 397 | 0.94 (0.74, 1.19) | 0.78 (0.49, 1.25) | 0.71 (0.50, 1.02) | 0.72 (0.59, 0.87) |
| Not drinking alcohol | 401 | 1.20 (1.00, 1.45) | 1.07 (0.79, 1.45) | 0.78 (0.55, 1.10) | 1.01 (0.85, 1.20) |
| Not smoking/using paan | 401 | 1.14 (0.95, 1.35) | 0.91 (0.66, 1.27) | 0.76 (0.55, 1.05) | 1.04 (0.89, 1.22) |
| Accuracy of iron-folic acid estimate | | | | | |
| Within 30 tablets | 402 | 0.96 (0.73, 1.28) | 1.06 (0.68, 1.68) | 1.17 (0.84, 1.63) | 1.01 (0.80, 1.29) |
| Within 60 tablets | 402 | 0.91 (0.75, 1.11) | 0.76 (0.50, 1.14) | 0.96 (0.76, 1.20) | 0.94 (0.81, 1.09) |
| Within 90 tablets | 402 | 1.00 (0.89, 1.12) | 0.69 (0.50, 0.96) | 0.96 (0.82, 1.13) | 0.99 (0.90, 1.10) |
| Estimate # of antenatal care visits | | | | | |
| Exactly | 400 | 0.67 (0.41, 1.10) | 0.47 (0.16, 1.40) | 1.25 (0.75, 2.08) | 0.94 (0.65, 1.36) |
| Within one visit | 400 | 0.86 (0.69, 1.08) | 0.87 (0.59, 1.27) | 1.07 (0.83, 1.37) | 1.04 (0.87, 1.24) |
| Within two visits | 400 | 0.85 (0.74, 0.99) | 0.76 (0.56, 1.04) | 1.07 (0.92, 1.24) | 0.93 (0.82, 1.04) |

*Adjusted risk ratio >1 indicates greater accuracy of reported services receipt.
†Adjusted for woman's age (continuous), any education, first pregnancy and socioeconomic status quartile.

to whether or not women had received specific nutrition interventions as a part of ANC visits. A likely explanation for null associations with the SDS is that such questions do not reflect practices that might be perceived negatively (or positively) from the standpoint of social norms.[25 26]

However, it is unclear why higher scores on the SDS were positively associated with more accurate recall of counselling on management of nausea. One possible explanation is that the counselling focused only on women that complained of nausea and not all women.[5] Thus, it is possible that the socially desirable response to this question was to accurately respond to whether or not counselling had been received and that these responses were particularly easy to recall because they were related to recollections of nausea.

In contrast to questions about the simple receipt of services during ANC visits, questions about the number of ANC visits or IFA consumption could be more subject to SDB because they relate to a woman's initiative in seeking services or taking supplements and because there is likely positive sentiment around the benefits of services for women and their babies. This could explain the finding that the presence of MILs and/or husbands during the interview was associated with a significantly greater overestimation of the number of ANC visits. Cognitive testing[27] conducted in this same population revealed that women were confused about whether visits to providers outside of health posts including for sonograms should be counted towards their estimate of ANC visits or not, and it is possible that women might be more likely to count such visits when in the presence of other household members.

IFA coverage is an important indicator for global monitoring of nutrition interventions; it is included in the Global Nutrition Monitoring Framework and an indicator for the Countdown to 2030.[28 29] There are global concerns about accuracy of recall of the number of tablets received or consumed during a specific pregnancy, particularly in the DHS or other surveys where respondents may be 2–5 years postpartum. Previous work in Nepal showed that at 6 months postpartum, women's self-reported receipt of IFA tended to overestimate the number of supplements that they actually received and that this overestimation was not associated with maternal characteristics including age, education, parity or socioeconomic status.[4]

Our study is the first that we are aware of to test a SDS in the context of rural Nepal, though SDSs have been used in urban India[30] and rural Bangladesh.[17] We pretested scale items and made some modifications. The reliability coefficient of our tested scale of 0.58 neared the threshold of 0.60 used to define acceptability and marginally surpassed that threshold when two items were removed. These values were slightly less than reliability coefficients documented in a study using a lengthier version of the Marlowe-Crowne SDS in four African countries, which varied from 0.63 (Kenya) to 0.80 (Ethiopia).[7] Many of the scale items in our study were positively correlated; however, it is uncertain how well the questions were understood by respondents, or the

extent to which the scale reflects social desirability in the context of rural Nepal. Additional work to develop and test questions that draw more strongly on local situations and norms would likely further strengthen the ability of the scale to capture social desirability in rural contexts, similar to what has been developed in the Philippines,[29] and could enhance the utility of such scales for use in surveys.

A strength of our study is that we used direct observation to assess the receipt of services in clinics. However, the tool used to assess social desirability has not been validated in a rural context and was only slightly adapted for comprehension by respondents in our area. One adaptation we made to improve comprehension by the study population was to reformat the SDS tool from statements into questions. Ultimately, however, it is not clear how well the scale was able to capture socially desirable response tendencies in this context. Our study used a recall period of approximately 6 months, which may have facilitated more accurate recall of receipt than typically is assessed in surveys such as the DHS, which currently administers ANC modules to women up to 2 years postpartum. The implications of different recall periods for SDB are unclear. Additionally, the enumerators noted the presence of others at any point in the interview and not specifically during the questions about receipt of services; without more detailed information about observer presence when the questions were asked, it is difficult to attribute findings specifically to SDB.

These findings reinforce the guidance often given during survey training to conduct surveys without others present, though doing so in practice is often challenging in rural village settings where norms around privacy differ. As we found that the SDS was not well correlated with overestimation or lack of accurate recall, there was no need to correct for SDB. Given that the generalisability of our study to other contexts including other parts of Nepal is uncertain, we recommend further studies be conducted in other contexts to develop and validate SDSs and use them to better understand the effects of SDB on estimates of services use.

**Acknowledgements** The authors thank the study team at the Nepal Nutrition Intervention Project-Sarlahi, who conducted household data collection. The Nepal Nutrition Intervention Project Sarlahi operates with the permission of and under the auspices of the Social Welfare Council in Nepal and in close conjunction with Nepal Netra Jyoti Singh. This work was funded by the Bill & Melinda Gates Foundation INV007332 and OPP1172551.

**Contributors** Conceived study design: MKM, JK, AT-L and RH. Implemented the study: TPL, SL, SKK, EB and PM. Helped interpret findings: all authors. Analysed the data and wrote the paper: AT-L. Read, provided edits and approved final manuscript: all authors. AT-L is responsible as guarantor for the study. All authors approve final content to be published and agree to be accountable for all aspects of the work.

**Funding** This work was funded by the Bill & Melinda Gates Foundation INV007332 and OPP1172551.

**Competing interests** None declared.

**Patient and public involvement** Patients and/or the public were not involved in the design, conduct, reporting or dissemination plans of this research.

**Patient consent for publication** Not applicable.

**Ethics approval** This study was approved by the institutional review board at Johns Hopkins Bloomberg School of Public Health (#8808) and by the Nepal Health Research Council (#441 2018). Participants gave informed consent to participate in the study.

**Provenance and peer review** Not commissioned; externally peer reviewed.

**Data availability statement** Data are available upon reasonable request.

**ORCID iDs**
Andrew L Thorne-Lyman http://orcid.org/0000-0001-5917-4126
Melinda K Munos http://orcid.org/0000-0002-1349-8026

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
