## [Reviewer comments · BMJ Open]

ARTICLE DETAILS

TITLE (PROVISIONAL)	How does social desirability bias influence survey-based estimates of the use of antenatal care in rural Nepal? A validation study
AUTHORS	Thorne-Lyman, Andrew; Lama, Tsering; Heidkamp, Rebecca; Munos, Melinda; Manandhar, Porcia; Khatry, Subarna; Bryce, Emily; LeClerq, Steven C.; Katz, Joanne

VERSION 1 – REVIEW

REVIEWER	Bergen, Nicole University of Ottawa
REVIEW RETURNED	07-Feb-2023

GENERAL COMMENTS	Congratulations on an excellent submission. This study was well designed and executed, and conforms to high reporting standards. The present manuscript has a few small inconsistencies to reconcile around the introduction and use of acronyms, as well as Supplementary Table numbering. The consideration of the presence of other adults during interviews was a valuable complement to the exploration of the SDS. Were your findings about the presence of interview observers novel? Have similar findings been reported elsewhere? Did you consider the potential role of other situational/interpersonal factors that might have affected social desirability bias, such as interview techniques, establishing rapport with the participant, presence of a male/female, local/foreigner interviewer? Were any strategies employed by the researchers to discourage (or inadvertently reinforce) social desirability bias and promote more accurate recall? In the discussion section (2nd paragraph), you raise the need for a better understanding of the perception of practices as negative/positive from the standpoint of social norms. To this end, did the researchers consult with qualitative evidence to gain insight into this? Or something that the research team/study staff could comment on, given their experience in the field? More generally, what existing knowledge is there from previous research on social desirability bias in Nepal (or rural Nepal)?
---

REVIEWER	Adhikari, Mukesh The University of North Carolina at Chapel Hill Gillings School of Global Public Health, Health Policy and Management
REVIEW RETURNED	17-Feb-2023

GENERAL COMMENTS	Given the higher reliance on national demographic health surveys in low and middle-income countries where the survey questionnaire can be subjected to social desirability bias (SDB),
--

	your study is highly relevant to address the concern of SDB during the data collection phase. The research is performed scientifically, and the manuscript is well-written. Here are my comments. I hope addressing these comments will make the manuscript better for publication. 1. Introduction: This section is clear and concise to show the literature gap and rationale of the study. One question is: you discussed 33 questions social desirability bias score, but your supplement table consists of 12 items. It's a bit confusing whether you applied 33 items or 12 items. If 12-item is the case, it needs further clarification, a bit in the introduction section and a detailed version in the method section. 2. Study Design: 2.1 In the measurement of social desirability, you discussed the rationale for 12 items and discussed internal consistency. Have you conducted a factor analysis to evaluate the contribution of items to the total variance? 3. Data Analysis: 3.1 Can you explain a bit about the reason to use the log-binomial bivariate model so that the readers can get a better sense? 3.2 Similarly, you mentioned the non-convergence issue to use log-poisson model models. It also needs more explanation so that readers can get a clearer idea. 4. Result: 4.1. Minor comment: Line 200: "Women with lower SDS tended to....." Is it higher or lower? The bottom quartile is 33.2% whereas the top quartile has 22.3%. 4.2. Could you show ethnic or caste variation in table 1? At the same time, have you accounted for ethnicity in the model? As the Terai region of Nepal is ethnically diverse, it would be interesting to see whether adjusting for ethnicity can vary your result. 4.3. Could you double-check this sentence? "Null relations were apparent between SDS and overestimation of receipt of most services, although many coefficients were positive when treating SDS as a dichotomous predictor". The risk ratio in the table includes 1 in the confidence interval while including SDS as a dichotomous predictor for all variables. 5. Discussion: 5.1. If possible, could you discuss more why assistance from MIL and husband are widely different for the accuracy of recall of services? There could be some cultural contexts in Terai regions. 5.2. "As we found that the social desirability scale was not well correlated with overestimation or lack of accurate recall, there was no need to correct for social desirability bias." I think you need to add " in this study", otherwise this seems a bit strong statement.
--	--

	Supplementary table 1: This table needs footnotes to clarify what the column actually means. In summary, your work is interesting and novel in the context of this topic and LMICs. I am excited to review the revised version.
--	---

REVIEWER	King , Bruce M Clemson University Department of Psychology
REVIEW RETURNED	12-Mar-2023

GENERAL COMMENTS	To me, there appears to be two major issues addressed by this study. First, was there evidence of social desirability biased responding? The answer is a very clear Yes by use of the gold standard -- observed overestimations when others were present. That is an important finding, and the authors correctly recommend that in the future these type questions be asked when no others are present. Second, because it will be impactable to always have observers present, the authors wished to determine whether the short version of the Marlowe-Crowne scale could be used to show social desirability biased responding. This proved not to be successful. However, the major problem with the study is the form of the M-C scale they used. They used the short form (13 questions) and then had to drop one of the questions (I like that they said that some of the questions "may not be sensitive from a social-norms perspective"). I have used the short form several times myself and was surprised at the wording of the 12 questions that were used. The authors appear to have reworded the questions. The M-C scale is a series of statements, not questions. Here is an example: Q.#12, authors' wording: Have you ever deliberately said something that hurt someone's feelings? Q.#12, original M-C wording: I have never deliberately said something that hurt someone's feelings. I do not know, nor can the authors know, how this change of wording might affect outcomes. The rewording in one case was completely wrong. For Q.#6, the authors end the sentence with "sometime?" The correct word is "someone." When I looked at Supplementary Table 1, I first thought that must be a typo, but it is repeated in Supplementary Table 3. That mistake renders this question useless, leaving just 11 questions (reworded) by which to calculate correlations. If this paper is to be published, the authors need to address the shortcomings of their M-C scale. The end conclusion for this aspect of the study is that this (flawed) version of the M-C scale is not useful for future studies of this type. It does not negate their important finding that social desirability biased responding occurred. A simple reorganization of the abstract would help. I suggest that the first sentence or two in the Results section of the Abstract state that social desirability responding as measured by the gold standard occurred, and then follow that with a statement that the short version of the M-C scale was not predictive of this. However, the authors should then indicate that there may have been problems with the scale itself.
--

REVIEWER	Henson, Laurence University of Washington
REVIEW RETURNED	19-Mar-2023

GENERAL COMMENTS	This a high-quality study that evaluates valuable global health efforts in resource limited areas. I applaud the authors in ensuring
--

	that the study was adequately powered, the thoroughness of its statistical methods, and attention to detail in comparing differences between different adults present during interviews. I only have minor suggestions/ questions, please see below:  1. Although this was closer to the end of the study period, how might have COVID precautions impacted survey results? The presence of masks, increased distance from the interviewer, etc. may have changed women’s need to respond in a socially desirable manner. I do not think it is necessary to perform another statistical test for this, however I think it is worth discussing. 2. Page 11, lines 174-175. While the selection of these variables appears internally consistent and lack signs of possible colliders, how were confounders determined for this study? Newer epidemiological evidence suggests that most published cohort studies in the past decades have lacked consistent confounder selection and proposed utilizing tools such as Directed Acyclic Graphs (DAG) for selecting covariates to adjust for in multivariate regression. Therefore, I recommend adding a brief rationale behind the authors’ selection of covariates. 3. Why do you suppose the bottom 3 quartiles had higher percentage of having another adult present during their interview? 4. Page 16, line 284. “Focused only on women that complained of nausea....”
--	--

VERSION 1 – AUTHOR RESPONSE

Reviewer 1	
Congratulations on an excellent submission. This study was well designed and executed, and conforms to high reporting standards.	Thank you very much for your kind words and suggestions to improve our paper.
The present manuscript has a few small inconsistencies to reconcile around the introduction and use of acronyms, as well as Supplementary Table numbering.	Thanks for catching this- we have searched for these and corrected those that we found.
The consideration of the presence of other adults during interviews was a valuable complement to the exploration of the SDS. Were your findings about the presence of interview observers novel? Have similar findings been reported elsewhere?	Yes, to our knowledge this was novel. We have clarified this in the introduction and discussion.
Did you consider the potential role of other situational/interpersonal factors that might have affected social desirability bias, such as interview techniques, establishing rapport with the participant, presence of a male/female, local/foreigner interviewer?	These are indeed important possibilities that could affect social desirability bias; as our goal was to assess the bias that might be present in normal health survey interviews, we tried to replicate these type of interviews as closely as possible (described below).
Were any strategies employed by the researchers to discourage (or inadvertently reinforce) social desirability bias and promote more accurate recall?	Interviewers were trained to replicate similar conditions that a normal field-based health survey might use. This included asking respondents to answer questions themselves even if a family member had tried to answer it for her and repeating/rephrasing questions in a

	simpler way if respondents were confused or responded with "I don't know". We've clarified this in the revised paper as well (lines 174-177).
In the discussion section (2nd paragraph), you raise the need for a better understanding of the perception of practices as negative/positive from the standpoint of social norms. To this end, did the researchers consult with qualitative evidence to gain insight into this? Or something that the research team/study staff could comment on, given their experience in the field? More generally, what existing knowledge is there from previous research on social desirability bias in Nepal (or rural Nepal)?	Thank you for this thought-provoking question. We could not find much literature on this from Nepal. Our sense, based mainly on our own perceptions and contextual knowledge from working in this setting for decades, is that two of the questions, pertaining to number of ANC visits and number of IFA supplements consumed could be more subject to social desirability bias than the others, given that (1) they relate to women's own initiative, and (2) that there likely is a positive sentiment around potential benefits of ANC services for women and their babies. In contrast, the other questions relate more to whether or not a service was given, and as such may be less subject to social desirability bias. We have expanded on in the discussion section.
Reviewer 2	
Given the higher reliance on national demographic health surveys in low and middle-income countries where the survey questionnaire can be subjected to social desirability bias (SDB), your study is highly relevant to address the concern of SDB during the data collection phase. The research is performed scientifically, and the manuscript is well-written. Here are my comments. I hope addressing these comments will make the manuscript better for publication.	Thank you for your helpful comments
1. Introduction: This section is clear and concise to show the literature gap and rationale of the study. One question is: you discussed 33 questions social desirability bias score, but your supplement table consists of 12 items. It's a bit confusing whether you applied 33 items or 12 items. If 12-item is the case, it needs further clarification, a bit in the introduction section and a detailed version in the method section.	We refer to the shorter questionnaire in our paper as "short-form" which is the standard terminology in the literature that uses the shorter tool.
2. Study design. 2.1 In the measurement of social desirability, you discussed the rationale for 12 items and discussed internal consistency. Have you conducted a factor analysis to evaluate the contribution of items to the total variance?	We did indeed conduct factor analysis to explore the dynamics of the social desirability scale but had decided not to present it given word limits and the desire to focus on the main findings rather than the tool itself.
3. Data Analysis: 3.1 Can you explain a bit about the reason to use the log-binomial bivariate model so that the readers can get a better sense? 3.2 Similarly, you mentioned the non-convergence issue to use log-poisson model models. It also needs more explanation so that readers can get a clearer idea	We've added an explanation to the former and a reference for readers with interest in seeing more about this method.
4.1. Minor comment: Line 200: "Women with lower SDS tended to....." Is it higher or lower? The bottom quartile is 33.2% whereas the top quartile has 22.3%.	Thanks for catching this we have fixed it.

4.2. Could you show ethnic or caste variation in table 1? At the same time, have you accounted for ethnicity in the model? As the Terai region of Nepal is ethnically diverse, it would be interesting to see whether adjusting for ethnicity can vary your result.	Yes it is true that the Terai is ethnically diverse and ethnicity presents a very interesting possible dimension to explore many things. Unfortunately as our sample came from a small number of clinics, with a more homogenous population, we do not have a diverse enough sample ethnically in this study to explore this.
4.3. Could you double-check this sentence? "Null relations were apparent between SDS and overestimation of receipt of most services, although many coefficients were positive when treating SDS as a dichotomous predictor". The risk ratio in the table includes 1 in the confidence interval while including SDS as a dichotomous predictor for all variables.	Thanks we have reworded for clarity. "Null relations were apparent between SDS and overestimation of receipt of most services, although some associations neared statistical significance when SDS was treated as a dichotomous predictor."
Discussion: 5.1. If possible, could you discuss more why assistance from MIL and husband are widely different for the accuracy of recall of services? There could be some cultural contexts in Terai regions.	Great question. We can only speculate on this finding- generally if the MIL lives with the nuclear family, she might be more influential on women and perhaps more privy to their day-to-day lives than their husbands, who may be away working, particularly for younger more inexperienced women and those with less education. However, this is just our speculation.
5.2. "As we found that the social desirability scale was not well correlated with overestimation or lack of accurate recall, there was no need to correct for social desirability bias." I think you need to add " in this study", otherwise this seems a bit strong statement.	Thank you we have added this
Supplementary table 1: This table needs footnotes to clarify what the column actually means.	Thank you we added a footnote explaining this
In summary, your work is interesting and novel in the context of this topic and LMICs. I am excited to review the revised version.	Thank you for the kind comment
Reviewer 3	
To me, there appears to be two major issues addressed by this study. First, was there evidence of social desirability biased responding? The answer is a very clear Yes by use of the gold standard -- observed overestimations when others were present. That is an important finding, and the authors correctly recommend that in the future these type questions be asked when no others are present.	Thank you for your helpful comments and suggestions. We have reflected on the interpretation you've given here- that the presence of others helped to predict the difference between reported and observed services use could be evidence of social desirability bias even if the social desirability scale itself did not explain this difference due to its potential limitations in this context. We agree that this could be an explanation, though we also feel that there may be other possible explanations at work here. One possible explanation that we can't rule out is residual confounding- perhaps common factors explain presence of others during the interview and the tendency to over report. We also don't have details about when during the interview others were present or what the nature of their interactions with the respondent were; we just know that they were present at any time during the interview. Without more information about

	such dynamics we are inclined to keep our original framing.
Second, because it will be impactable to always have observers present, the authors wished to determine whether the short version of the Marlowe-Crowne scale could be used to show social desirability biased responding. This proved not to be successful. However, the major problem with the study is the form of the M-C scale they used. They used the short form (13 questions) and then had to drop one of the questions (I like that they said that some of the questions "may not be sensitive from a social-norms perspective"). I have used the short form several times myself and was surprised at the wording of the 12 questions that were used. The authors appear to have reworded the questions. The M-C scale is a series of statements, not questions. Here is an example:	One of the challenges of applying tools across different languages and cultural contexts is differences in interpretation, as noted in the discussion section. While pretesting the questions in our study site (a rural setting), we found that the format of asking women to agree or disagree with a statement was not a familiar or understandable approach by the respondents. So, we transformed it into a question format that enabled better comprehension by respondents. We agree that this results in uncertain changes in meaning but given the tradeoff of comprehension it seemed to us to be a reasonable compromise. We have elaborated on this more in the methods and study limitations section.
Q.#12, authors' wording: Have you ever deliberately said something that hurt someone's feelings? Q.#12, original M-C wording: I have never deliberately said something that hurt someone's feelings. I do not know, nor can the authors know, how this change of wording might affect outcomes. The rewording in one case was completely wrong.	It is true that it's not possible to know how word changes affect questionnaire meaning, but at the same time this is an inevitable occurrence when translating tools across languages and cultures, something that we have acknowledged as a limitation in the discussion section. As noted below, there was a typo which is now corrected.
For Q.#6, the authors end the sentence with "sometime?" The correct word is "someone." When I looked at Supplementary Table 1, I first thought that must be a typo, but it is repeated in Supplementary Table 3. That mistake renders this question useless, leaving just 11 questions (reworded) by which to calculate correlations. If this paper is to be published, the authors need to address the shortcomings of their M-C scale.	Thank you for catching this- it was indeed a typo in both places. The administered tool used the correct word "someone". Accordingly, we have retained all the items. In the paper limitations section of the discussion we have addressed issues related to the need for more work to locally adapt and validate this and other social desirability scales.
The end conclusion for this aspect of the study is that this (flawed) version of the M-C scale is not useful for future studies of this type. It does not negate their important finding that social desirability biased responding occurred. A simple reorganization of the abstract would help. I suggest that the first sentence or two in the Results section of the Abstract state that social desirability responding as measured by the gold standard occurred, and then follow that with a statement that the short version of the M-C scale was not predictive of this. However, the authors should then indicate that there may have been problems with the scale itself.	Thank you for this suggestion on framing, which we have considered and addressed above.
Reviewer 4	
This a high-quality study that evaluates valuable global health efforts in resource limited areas. I applaud the authors in ensuring that the study was adequately powered, the thoroughness of its statistical methods, and attention to detail in comparing differences between different adults	Thank you for your kind words and helpful suggestions.

present during interviews. I only have minor suggestions/ questions, please see below:	
1. Although this was closer to the end of the study period, how might have COVID precautions impacted survey results? The presence of masks, increased distance from the interviewer, etc. may have changed women's need to respond in a socially desirable manner. I do not think it is necessary to perform another statistical test for this, however I think it is worth discussing.	We do not think the precautions influenced the interviewing dynamics very much, but this is our anecdotal opinion.
2. Page 11, lines 174-175. While the selection of these variables appears internally consistent and lack signs of possible colliders, how were confounders determined for this study? Newer epidemiological evidence suggests that most published cohort studies in the past decades have lacked consistent confounder selection and proposed utilizing tools such as Directed Acyclic Graphs (DAG) for selecting covariates to adjust for in multivariate regression. Therefore, I recommend adding a brief rationale behind the authors' selection of covariates.	Yes we also come from an epidemiology background and use theory-based selection. We've added this to the paper.
3. Why do you suppose the bottom 3 quartiles had higher percentage of having another adult present during their interview?	Great question. The presence of others, particularly mothers in law or other adults may have influenced responses to the social desirability module, making them less likely to agree with social desirable statements. Of course, we do not know whether observers were present for that section or not and there could be other explanations...
4. Page 16, line 284. "Focused only on women that complained of nausea...."	Thank you we have fixed the typo

VERSION 2 – REVIEW

REVIEWER	King , Bruce M Clemson University Department of Psychology
REVIEW RETURNED	25-Jun-2023
GENERAL COMMENTS	I was somewhat critical about the short-form M-C scale in my first review, but I feel that the authors have made a good faith revision. Congratulations!
REVIEWER	Henson, Laurence University of Washington
REVIEW RETURNED	27-Jun-2023
GENERAL COMMENTS	No further recommendations at this time, all my previous queries were already addressed by the authors. Excellent work.

VERSION 2 – AUTHOR RESPONSE